# Predicting the Structure and Dynamics of Membrane Protein GerAB from *Bacillus subtilis*

**DOI:** 10.3390/ijms22073793

**Published:** 2021-04-06

**Authors:** Sophie Blinker, Jocelyne Vreede, Peter Setlow, Stanley Brul

**Affiliations:** 1Molecular Microbiology and Food Safety, Swammerdam Institute for Life Sciences, University of Amsterdam, 1098 XH Amsterdam, The Netherlands; sophieblinker@hotmail.com; 2Computational Chemistry, Van’t Hoff Institute for Molecular Sciences, University of Amsterdam, 1098 XH Amsterdam, The Netherlands; 3Department of Molecular Biology and Biophysics, UConn Health, Farmington, CT 06030-3305, USA; setlow@uchc.edu

**Keywords:** molecular dynamics, *Bacillus subtilis*, spores, germination, germinant receptor

## Abstract

*Bacillus subtilis* forms dormant spores upon nutrient depletion. Germinant receptors (GRs) in spore’s inner membrane respond to ligands such as L-alanine, and trigger spore germination. In *B. subtilis* spores, GerA is the major GR, and has three subunits, GerAA, GerAB, and GerAC. L-Alanine activation of GerA requires all three subunits, but which binds L-alanine is unknown. To date, how GRs trigger germination is unknown, in particular due to lack of detailed structural information about B subunits. Using homology modelling with molecular dynamics (MD) simulations, we present structural predictions for the integral membrane protein GerAB. These predictions indicate that GerAB is an α-helical transmembrane protein containing a water channel. The MD simulations with free L-alanine show that alanine binds transiently to specific sites on GerAB. These results provide a starting point for unraveling the mechanism of L-alanine mediated signaling by GerAB, which may facilitate early events in spore germination.

## 1. Introduction

Germination of spores of the bacterium *Bacillus subtilis* can be induced by exposure to a variety of sugars, amino acids and cationic ions [1,2]. Initiation of germination is presumably mediated by a protein complex located in the spores’ inner membrane (IM) called the germinosome [1,3,4,5]). The germinosome is composed of the GerD protein and three Germinant Receptor (GRs), A, B and K, each of which contain A, B and C subunits. Clustering of GRs in a germinosome appears to greatly increase GR function and downstream signaling which eventually lead to spore germination and outgrowth [1]. Different GRs respond to different nutrients called germinants, and the amino acid L-alanine alone can induce germination via the GerA GR. While there is no direct evidence for L-alanine binding to any GerA subunit, mutagenesis experiments suggest that germinants can bind to either the GerAA or GerAB subunits, and perhaps binding occurs to both subunits, either cooperatively or sequentially [6,7,8,9,10]. Other mutagenesis experiments indicate that GerAA and GerAB interact [11]. High-resolution structures of GR A and C subunits have been determined by structural and functional analysis [4,9,10,12,13,14], but for the B subunit no structural information exists. Based on amino acid sequence, the B subunit is most likely an integral IM protein with multiple transmembrane helices [1]. The current view is that once GRs sense germinants such as L-alanine, they undergo a conformational change, most likely in at least a region of a GR subunit in spores’ IM, which results in increased IM permeability to small molecules [1,3,15]. This latter change commits a spore to germinate and leads first to the intake of some water and the release of core monovalent cations and small amounts of the core’s huge pool of Ca-dipicolinic acid (CaDPA), which constitute 25% of spore dry core. This is followed by rapid release of all CaDPA in a few minutes and more water uptake, raising core water content from 35% of wet weight in the dormant spore to 45%. In turn, this latter event triggers hydrolysis of spore cortex peptidoglycan which allows core swelling and water uptake, giving a germinated spore with a core water content of 80% which is similar to that in growing cells.

Spore formers including *B. subtilis* are associated with food spoilage and food borne diseases [16], in large part because these spores can remain dormant and resist environmental stresses for a very long time [17]. However, when spores germinate, they lose their resistance properties and become relatively easy to kill, although rates of germination of individual spores in populations are very heterogeneous [1]. The loss of spore resistance in germination has led to the so called “germinate to eradicate” strategy for spore decontamination. Consequently, it would be advantageous to broaden our knowledge of the germination process, in particular the precise mechanism by which spore germination is initiated via GRs. As noted above, it appears most likely that IM embedded regions of GR proteins, with the GR B subunits the major IM-embedded GR subunit, are intimately involved in the very beginning of germination, in particular the binding of a germinant and then the transduction of the signal triggering initiation of germination. Thus, any new information about IM regions of GR subunits could provide new insight on GR-dependent spore germination. The *B. subtilis* GerAB protein and its homologs in other GRs, belong to the superfamily of amino acid polyamine organocation (APC) secondary transport proteins [18]. GerAB has 365 amino acids and is predicted to contain 11 transmembrane (TM) domains by SAM (Sequence Analysis Methods for automatic annotation of unreviewed entries), using a combination of TMHMM and Phobius [19,20,21,22], see Uniprot entry UniProtKB - P07869 (GERAB_BACSU). However, a study of the membrane topology of GR proteins in *B. anthracis*, suggested that the B subunit consists of 10 TMs [13]. In contrast to the APC transporters which move substrates into growing cells from the outside, GerAB and homologs seem much more likely to function as transceptors, to which bound ligands trigger downstream signaling [18,23]. Notably, it has been suggested that *B. subtilis* GerAB exhibits structural similarity to several types of proton coupled broad specificity amino acid transporters, ApcTs and LeuT [6]. Both ApcTs and LeuT represent one group of APC transporters, have a similar cylindrical structure [24,25,26], and have amino acid ligands, as *B. subtilis* GerAB may also have. However, unlike ApcTs and LeuTs, little is known about the ligand binding pocket in GerAB. Cooper and Moir, (2011) attempted to determine residues involved in L-alanine binding by a mutational study on GerAB [6]. They showed that L-alanine germination was significantly slowed by mutations in GerAB residues L24, E51, P178, T192, and R271, although the residues that might form the ligand binding pocket were not identified. However, support for at least some of the mutated residues being implicated in ligand binding was given by Edwards et al. (2018) who argued that amino acids occupying the equivalent position of V104 in LeuT fold are essential for substrate specificity [27]. In addition, LeuTaa, part of the APC subfamily NSS, showed a ligand binding site comprising residues from helix 1, 6 and 8. The binding pocket of LeuTaa consists of hydrophobic as well as hydrophilic interactions [24]. Moreover, the GkApcT amino acid transporter has a binding site for L-alanine close to residues I40, T43, G44, F231, A232 and I234 which are also located in helices 1 and 6 [26]. Jungnickel et al. (2018) showed that the structure of GkApcT adopts a pseudo two-fold symmetry axis in which TM1-TM5 are related to helices TM6-TM10.

The function of GerAB is almost certainly affected by its fold but may also be influenced by interactions with other GR subunits, as well as with other proteins in the germinosome. Since GerAB is very likely to be bound to the IM of the spore, both the membrane composition and structure may affect GerAB function. Fatty acids in the IM of *B. subtilis* spores are predominantly 16–18 carbon saturated fatty acids, very similar to the fatty acids in the growing cell plasma membrane [28]. The IM of *B. subtilis* spores is mainly composed of phosphatidyl glycerol (PG, 63%), cardiolipin (CL, 27%) and phosphatidylethanolamine (PE, 10%) [29]. The absence of CL from the spore IM was initially reported to almost eliminate spore germination [29], but a later study found only an 30% slowing of the germination rate in CL-less spores [30]. It is also important to note that the IM in intact *Bacillus* spores may have a slightly altered structure than that of the plasma membrane of growing cells, since: (i) lipids in the IM have an extremely low mobility, much lower than that in growing cells or germinated spores plasma membrane; and (ii) the IM permeability is much lower to small uncharged molecules, including water, than are plasma membranes of growing cells or germinated spores [31,32].

As no structural data is available for GerAB, it is unknown which residues are involved in L-alanine binding. We therefore conducted a Molecular Dynamics (MD) study on *B. subtilis* GerAB, aiming to elucidate the structure and identify possible L-alanine binding sites. Such insights will deepen our knowledge of germination proteins in general.

By first constructing initial structures using a comparative modeling approach, and subsequently equilibrating these models embedded in a membrane, this study presents a qualitative structural prediction for GerAB. By performing MD simulations in the presence of free L-alanine, we were able to identify potential L-alanine binding sites. Note that the MD simulations presented here are by no means converged, and we can therefore only present qualitative results. Quantitative predictions such as free energy of binding, affinities etc. are outside the scope of this work. This work shows that the structure of GerAB is predicted to be similar to other ApcT and LeuT transporters. GerAB also contains a water channel that traverses the membrane spanning portion which might be involved in transporting molecules as well as water across the IM. This study provides a starting point for further research into spore germination proteins.

## 2. Results

### 2.1. Structure of GerAB

Structural models for GerAB were constructed based on similarities between GerAB and ApcT transporters. With the aid of the online tool SWISS-model, GerAB was aligned with potential templates, using a template based approach [33]. The SWISS-model search resulted in the first potential template to model the structure of GerAB, GkApcT (PDB code 6F34) [26]. 6F34 was identified as the best suitable template because the model showed the best QMEAN score [34] (13.99% sequence identity, sequence similarity of 0.28, QMEAN = −5.57). Previously, amino acid sequence comparisons suggested that GerAB shows similarity with that particular group of transmembrane proteins [6]. Even though 6F34 has a longer amino acid sequence than GerAB, 461 amino acids, only 303 residues could be aligned since the similarity of the last part was too low. Therefore, we turned to a second web-based tool RaptorX [35]. Using the template based approach in RaptorX we found a second template that could fully align GerAB, resulting in a structure with QMEAN = −8.49. 3GI8 was the second hit in RaptorX for GerAB but also showed the highest sequence similarity in SWISS-model (16.38% sequence identity, sequence similarity 0.28, QMEAN = −6.65). However, again SWISS-model could not fully align GerAB to 3GI8 [33]. For that reason two 3GI8 models were created: one with SWISS-model and one with RaptorX, respectively 3GI8-SWISS and 3GI8-RaptorX. 3GI8 is also a ApcT transporter ApcT K158A (PDB code 3GI8) [25], similar to 6F34. Figure 1 shows the sequence alignments of GerAB with the templates. To summarize: we created three models: 3GI8-SWISS, 3GI8-RaptorX and 6F34-SWISS. The pairwise RMSD as calculated for the backbone atoms in the residues present in all models is below 0.3 nm: RMSD${}_{3\mathrm{GI}8-\mathrm{RaptorX}-3\mathrm{GI}8-\mathrm{SWISS}}$ = 0.249 nm, RMSD${}_{3\mathrm{GI}8-\mathrm{RaptorX}-6F43-\mathrm{SWISS}}$ = 0.244 nm and RMSD${}_{3\mathrm{GI}8-\mathrm{SWISS}-6F43-\mathrm{SWISS}}$ = 0.274 nm.

While inserting GerAB into a membrane, the N-terminus was trimmed by 3 and 6 residues to minimize a long unstructured terminus, for the 3GI8-SWISS and 6F34-SWISS models respectively, resulting in a total length of 300 amino acids for 3GI8 and 297 amino acids for 6F34. For 3GI8-RaptorX, no N-terminal processing was needed during insertion in the membrane. After inserting the structural models into a membrane consisting of palmitoyloleoylphosphatidylglycerol (POPG), cardiolipin (TMCL2) and palmitoyloleoylphosphatidylethanolamine (POPE), we subjected the GerAB models to 5 runs of 100 ns MD simulations to test the stability of the models. The GerAB models were similarly checked in presence of 36 mM L-alanine (corresponding to 20 molecules) at ambient conditions. The high L-alanine concentration was used to speed up any association processes to achieve an interaction between alanine and the protein within 100 ns. As a control, simulations of GerAB with 1 molecule of L-alanine, representing a concentration of 1.8 mM, demonstrated that the structure of the protein is not affected by the presence of higher ligand concentration. Visual inspection showed that all three structural models stay intact during the simulations.

In all three models the transmembrane α-helical bundle is preserved. Moreover, GerAB remains stable in the membrane. Figure 2 shows a schematic overview of the topology of secondary structure elements in GerAB, based on the simulations of the three structural models. Figure 2A shows the protein as a flat structure, Figure 2B illustrates the organisation of the protein from the inside of the IM. Figure 3A–C shows snapshots of GerAB after 100 ns of molecular dynamics for respectively 3GI8-SWISS, 6F34-SWISS and 3GI8-RaptorX. Both the amino- and the carboxy terminus are located on the same side of the membrane, which can therefore be labeled as outside [13]. The structural model of GerAB contains transmembrane helices (TMs), two helices on the inside of the membrane (intracellular loop, ILs) and one helix on the outside of the membrane (extracellular loop, EL).

Most TM helices are oriented perpendicular with respect to the membrane. The first helix TM1 is located in the middle of the protein and is flanked by TM2, 3, 6, 7 and 8. The TM2 segment is a long helix on the outside of the protein. TM3 is a helix, oriented in a tilted conformation with respect to the membrane. TM8 is also a helix, in a similar orientation as TM3. Helices TM4 and 5 form a V-shaped structure, with the bottom of the V at the outer side of the membrane. TM4 and 5 interact with TM3 and 8 in an orientation reminiscent of a pincer. The sixth TM region is located on the inside of the protein, enclosed by TM1, 2, 3, 7, TM 8 and has an anti-parallel orientation with respect to helix 1. Unlike the other TMs, TM6 has a large extended non-helical region with only a small helical conformation at its beginning at the outer side of the membrane. Helices TM9 and 10 also form a V-shaped structure, in opposite orientation to TM4 and 5, with a smaller distance on the inner side of the membrane. GerAB also contains helical regions in the loops connecting the TMs. The helix in extra-cellular loop EL1 is 7 amino acids long, and in the reverse orientation to helix 3. On the intra-cellular side two loops contain helices, IL3 and IL4. The loop IL3 contains a helix of 1-2 turns long. IL4 contains two separate small helices, IL4a and IL4b, both 6 amino acids long. Overall, the structural models as constructed in this work contain many similarities.

When comparing the three structural models, we observed several differences. Most apparent is that the 3GI8-RaptorX model contains 10 TM regions, while 3GI8-SWISS and 6F34-SWISS contain only 8, due to the incomplete alignment of 3GI8-SWISS and 6F34-SWISS. Helix 9 and 10 form an upside-down V over helix 3, resembling the structure of TM4 and 5. 3GI8-SWISS and 6F34-SWISS have 3 ELs and 4 ILs, while 3GI8-RaptorX contains an extra loop region on each side of the membrane. The length of the helices varies between the models, in particular, with differences in TM1, TM3, TM4, TM6 and TM8. TM helix 1 is continuous in 3GI8-SWISS and 3GI8-RaptorX, but in 6F34-SWISS this region contains a disruption in the helical conformation at its center. In 3GI8-RaptorX, TM4 shows a more inward facing orientation compared to the other models. There is a difference to what extent TM8 and TM3 traverse the V-shape created by TM4 and TM5. This might also be affected by the length of the helices. For instance, in the 3GI8-RaptorX model TM3 and TM8 contain longer helices compared to the 3GI8-SWISS and 6F34-SWISS models, resulting in a larger portion traversing the TM4-5 V-shape. TM6 has the least secondary structure in 3GI8-RaptorX, while in 6F34-SWISS, this region contains two small helices. TM8 shows a disruption in 6F34-SWISS but is continuous in the other models.

The connecting loops vary more both in number of amino acids, orientation and structure. For EL1 the difference concerns the orientation towards helix 3. In addition, IL3 in 3GI8-RaptorX is longer compared to the other models and lacks a small helix. In 3GI8-SWISS and 3GI8-RaptorX IL4 contains 2 α-helical regions of 7 residues in a perpendicular orientation to each other. In 6F34-SWISS IL4b is missing (Figure A2C). At the end of the simulations, the intra- and extra-cellular loops seem longer and less organised defined by the length of the loop and the portion of secondary structure. The helix in EL1 has a different orientation with respect to helix 3 in each model. In total, 5 runs for each model, starting from different initial velocities, were performed (Table A1) to study the dynamics of GerAB in a membrane, resulting in similar structures (Figure A2).

The RMSD was calculated for all atoms in the protein with respect to the starting structure (i.e. the energy minimized model after embedding in the membrane) for every run and show that after 50 ns, the RMSD remains constant, indicating that by then, the systems are equilibrated. Overall 3GI8-RaptorX shows the highest RMSD (max RMSD 0.6 nm) and 6F34-SWISS the least (max RMSD 0.4nm) (Figure A1). In further analysis, we have used the last 50 ns of the MD simulations. We also compared the structural models to their templates (Figure A2). The 3GI8-SWISS and 6F34-SWISS models lack four helices in comparison to their templates while 3GI8-RaptorX misses only 2 helices. In addition, TM3 in the template and 3GI8-RaptorX model is more buried on the inside of the protein while in the 3GI8-SWISS and 6F34-SWISS models TM3 is primarily located on the outside. Differences occurred primarily in the orientation of the loops. In general, the template structure is conserved in the models.

For a more in-depth analysis of the helical structure of GerAB we calculated the helicity. The helicity is a measure of α-helical content based on the distance between the hydrogen bond acceptor C=O in residue n and the hydrogen bond donor N-H in residue n+4. When this distance is below 0.35 nm a hydrogen bond is formed. The red blocks in the graphs of Figure 3A–C indicate TM helices as predicted by a combination of two transmembrane helix prediction algorithms TMHMM (transmembrane hidden Markov model) and Phobius [19,20] in Uniprot [36] entry UniProtKB-P07869 (GERAB_BACSU). The helicity plots of 3GI8-SWISS and 6F34-SWISS show 7 regions where the distance is primarily under 0.35 nm and for 3GI8-RaptorX 9 such regions can be observed. This is one helix less than predicted by TMHMM/Phobius, because TM 9 could not clearly be distinguished from the helicity data. Remarkably, the helicity at the predicted helical structure for TM8 (8th red block in the graphs) shows values between 0.6–0.9 nm, indicating that no (TM) helix is formed here. Further structural analysis revealed that the TMHMM/Phobius predicted TM8 is in fact IL4, containing either one or two small helices. TM helix 1 contains less helical character in the middle of the sequence which is in agreement with the structural data indicating lower helicity in that in that area. A small dip in the helicity after helix 2 represents the small helix of EL1. High values for the helicity occur for the region after helix 5 up to helix 7. This region contains IL3 and TM6. The structures in Figure 3 reveal that TM6 does not fully contain an α-helical structure.

In addition, the helicity plots indicate that the lengths of the helices vary slightly between the models, in particular for helix 4. Although TM9 and 10 are easily deduced from the structure in Figure 3, the helices are not obvious in the helicity plot. For the precise residues, as deduced from the helicity data, that form the TMs see Table A1.

Overall, the predicted helices coincide with the observed helices except for predicted helix 8. However, TM6 is neither deduced from the structural analysis nor from the helicity plots.

### 2.2. Interaction with Membrane, Water and Ions

To identify residues that are in contact with water and or the membrane, we first calculated the minimum distance of a residue to water molecules, the aliphatic carbon atoms in the lipid tails or K${}^{+}$ ions. The proximity is then defined as the probability of finding any of these particles within 0.4 nm of a residue. A proximity of 1 indicates that a residue is always interacting with a particular molecule type and a proximity of 0 implies that no interaction occurs at all. Calculations were performed on the last 50 ns of 5 runs for each of the models.

In Figure 4B the proximity of lipid tails to protein residues is illustrated. This graph shows that most residues interact with membrane tails but no clear specific interacting regions can be identified. The proximity of the tails is mostly the opposite of the proximity of water, as expected. Even though regions that solely interact with membrane tails are less easily deduced from this graph, it does show regions of residues that have a very low proximity with tails (e.g., residues V19-P42, Q66-W85, F198-F218, V239-E270, T287-T301). Residues V19-P42 correspond to TM1 and residues F198-F218 are part of TM6, and thus a low proximity to tails can be explained by their position which is in the middle of the protein. On the other hand, residues V239-E270 belong to IL4 which is presumably in closer contact with water molecules. Conversely, residues with an higher proximity to the membrane tails (e.g., K9-I18, W45-I65, I86-A108, T121-I170, G183-G200, R219-I238, R271-T286), are located in TM domains of the helices.

Figure 4C shows the proximity of water to residues in GerAB. Residues around A20, L50, G100, R120, P150, I230, W280 interact little with water while residues M1-F13, T30-T47, T63-I89, A108-V125, V139-P150, K169-Y220 and G241-F275 are always exposed to water (Proximity = 1). Low hydration is found between residue I90 and R107 which are part of TM3. Residues A221 to V240 belong to TM7 which also shows low hydration indicating that this residue is more buried from water molecules. Residues I196-M201, in TM6, show more varying values for the proximity, indicating partial hydration. This might be the result of slower or less water entering in time. High hydration is observed for ILs and ELs such as residue K169-I187 for IL3 and G241-F275 which represents IL4.

In the center plane of the protein mainly residues of TM6 are interacting with water while TM1, 2, 3 and 8 residues show interaction with water to a lesser degree (proximity of water higher than 0.95: residues of 3GI8-RaptorX TM1 T22, M23, G25, A26; TM2 F55; TM3 Y97, F98, F104, E105; TM6 I196, F198-M201; TM8 T287. Residues of 3GI8-SWISS TM3 A102; TM6 S195, S197. Residues of 6F34-SWISS TM3 E105; TM6 V194, S195). The overall similarity in patterns of proximity between the models indicates that the residues that interact with water remain hydrated for the duration of the simulations. Figure 4D provides an overview of the interaction of membrane in white and water in blue with GerAB. It shows that the protein is located with its center plane in the middle of the membrane (Figure 2). Also, this visualisation shows the shape of the membrane around the protein.

### 2.3. GerAB Harbours a Water Channel

It was expected that a binding pocket for L-alanine would reside on the inside of the protein (TM1 and 6). In order to deliver a ligand to this position, this site should be accessible to water. Figure 4D already showed that water is not only located on the outside of the membrane. This channel is comparable in the three different models. Residues lining this channel (obtained from proximity data and HOLE analysis [37]) comprise residues of TM1, 2, 3, 6 and 8 (including residue S17, I18, N21, T22, A26, F55, F58, V101, F104, S197, T287, I290 and Y291). In addition, we computed the radius of the channel as a function of the position along the axis perpendicular to the membrane, using the procedure as outlined in Ref. [37] for frames extracted every 200 ps. (Figure A3). Openings of radius 0.1–0.3 nm are formed in the protein in all simulations. These results show that water permeates the protein from below in all simulations. A water channel even spans the membrane for for runs 2, 4 and 5 of 3GI8-RaptorX and run 1 of 6F34-SWISS. Water is not present inside the protein at the start of the simulations and must have entered during the simulations. We visualized the formation of the water channel in a video provided as online Appendix A.

To examine this process more thoroughly, we determined the number of water binding events for each residue. A binding event is defined as follows: if in one frame in a simulation run there is no water molecule within 0.4 nm of a residue, followed by a frame where a molecule has come within this distance, the second frame is counted as a binding event. Counting the time after a binding event until the distance between the residue and water has become larger than 0.4 nm will give the residence time. This can be averaged over all binding events (Nb) for each system, resulting in the average residence time, $\langle t\rangle$. These quantities are shown in Figure 5 for each system. A high number of binding events is generally observed for residues in the C-terminus, around residue G100 (TM3), I230(TM7) and K265 (IL4) (Figure 5A).

The first graph of Figure 4 presents the proximity of potassium to GerAB (Figure 4A). Peaks are related to proximity with water (Figure 4C) where residues that are exposed to water are also more often in contact with potassium. Especially residues near R120 (begin TM4), G140 (begin TM5), L180 (IL3) and E270 (begin TM8) show high proximity with potassium. The three models are comparable and differ only slightly in the height of the peaks. See Figure A4 for the average residence time and number of binding events for K^+^. We did not observe any potassium going into the opening formed inside the protein.

Mutation of residue E51 (TM2) reduces germination efficiency, suggesting this residue is of functional importance (see also Discussion) [6]. In addition, Edwards et al. (2018) proposed that equivalents of V104 in PAT2, an amino acid transporter of the APC superfamily, are associated with the binding pocket of a ligand [27]. In the case of GerAB this would be V101 (TM3). Residues L31, S195 and M201 are also believed to be involved in ligand binding [24,25,38]. These residues are part of TM1, 3 and 6. Proximity data shows that residue L31 has a proximity to water higher than 0.99 and an average residence time of respectively 0.02 and 0.2 ns for 3GI8-SWISS and 3GI8-6F34. A proximity of 0.86 is observed for residue F101 in 3GI8-Raptorx with an additionally low average residence time (Figure 5B). Conversely, residue V104 shows a high average residence time in 3GI8-RaptorX. Residue S195 and M201 show high proximity to water, a moderate number of binding events but a low average residence time. High average residence times for channel residues are observed for residues F104 (0.25 ns) Y291 (0.01 ns) and F104 in 3GI8-RaptorX (Figure 5B).

Based on the results using HOLE, the proximity data, the number of binding events and the average residence time we observed that most likely a water channel is present in GerAB.

### 2.4. Interaction of GerAB with L-Alanine

The MD simulations of 3GI8-SWISS, 6F34-SWISS and 3GI8-RaptorX contained excess L-alanine, for which we determined the average proximity over 5 runs per model between L-alanine and GerAB residues. The plot in panels A-C in Figure 6 shows that L-alanine is more often located to certain residues. The different models show similar peaks for L-alanine proximity and vary only in height. It appears that the proximity of L-alanine is higher around selected residues as follows. With a proximity of higher than 0.1, L-alanine was often associated with residues K68, E80, R177 and K215 in the 3GI8-RaptorX model. Similar results were obtained for the 3GI8-SWISS model with a proximity of higher than 0.1 for residues Y216 and Y220. Proximity of L-alanine to 6F34-SWISS was high for residues T121, K169, N174, P178, K213, K214, T217, R271 and F272. In all three models residues with a proximity above 0.1 are present around R177 and K215. Both residues are located in connecting loops with respectively the former in IL3 and the latter in EL3. Peaks of L-alanine proximity coincide with high proximity of water and potassium. To better illustrate the residues with a high proximity to L-alanine, residues in Figure 6D–F are colored according to their proximity with L-alanine ranging from lower (P${}_{Ala}$ of 0.01–0.3) in purple blue to higher (P${}_{Ala}$ of 0.4–1) in pink. From the figure it becomes clear that L-alanine favours residues located in the connecting loops. Colored areas in 3GI8-RaptorX indicate residues R177 and L215, in 3GI8-SWISS R219 and L215 are highlighted and in 6F34-SWISS I173, R177, L212, L213 and T216 are colored.

To clarify in what fashion L-alanine approaches GerAB, the number of binding events and the average residence time have been determined (Figure 6G–I). A L-alanine binding event occurs when L-alanine comes within 0.4 nm of a residue for the first time after being at a larger distance. Multiple binding events can occur within 100 ns. For each event, we counted the time the molecule stays within 0.4 nm. This residence time is then averaged over the number of binding events (Nb) to give the average residence time, $\langle t\rangle$. A high number of binding events combined with a low average residence time indicates that the molecule binds only transiently, which a low Nb combined with a high $\langle t\rangle$ indicates a strong interaction. In Figure 6J–M the number of binding events is shown. For 3GI8-RaptorX, the highest number of events (except the N and C-terminus) is observed for residue E80 (1197) and pairs with 27 ps of average residence time. Conversely, residue P186 shows a number of 1094 events but only 4 ps average residence time is recorded and for residue G185 75 events occurred with an average time of 207 ps, see Figure 6J. The 3GI8-SWISS model shows high numbers of L-alanine binding events for residues H69, G81, R120, G183 and T216. Whereas only a specifically a high average residence time was observed for residue Y220 (488 ps), with 94 events (Figure 6K). The last plot of Figure 6 shows the number of binding events for L-alanine to 6F34-SWISS. Here, residues 165 (IL3) and 210 (EL3), with respectively 1341 and 1440 events, show two higher peaks. Yet, this corresponds to an average residence time of 8 and 12 ps. Other noticeable peaks in events with their corresponding average residence time are observed for residues L38 (19 ps), H69 (18 ps), R120 (30 ps) N174 (37 ps), F272 (36 ps) (Figure 6L). Overall average residence time in the three models is higher than 50 ps around residues N40 (EL1), L180 (IL3), K215 (EL3) and T250 (IL4).

## 3. Discussion

In this study we present a prediction of the structure of GerAB by MD based on similarity with GkApcT and ApcT K158A at atomic detail. To accomplish this, three models were evaluated.

Our results indicate that GerAB is a transmembrane helical bundle that presumably consists of 10 TM helices which adopt a pseudo two-fold symmetry axis. While most TMs showed a helical conformation, TM6 partly loses its α-helical conformation during relaxation of our initial structural models in a membrane. This unwinding of TM6 may be required to allow L-alanine to enter and interact with GerAB. Another explanation could be that TM6 is not stable as an α-helix. EL1 is located close to TM6 and shows a wide variety of orientations. If water enters the inside of the protein from the extracellular side, then EL1 may function as a sort of gatekeeper where an orientation more anti-parallel to TM3 allows water to more easily flow in.

The proximity data for water and membrane tails revealed a number of regions that manifest a proximity to both water and membrane tails that is not exclusively high or low. This is especially true for residues in TM1 and 6 and can be explained by the hydration of residues that line the water channel which primarily are part of TM1 and 6. A more in-depth examination of the water channel revealed that the residues in the channel participate in binding with water molecules (Figure 4 and Figure 5). The function of GerAB may therefore be twofold, by on the one hand acting as a transceptor for L-alanine and on the other hand transporting water to the inside of the spore.

In searching for a potential binding pocket for L-alanine, we performed MD simulations of GerAB embedded in a membrane at much higher concentrations than used in experiments [39]. Despite the much higher concentration of L-alanine, only one simulation showed interactions between protein and ligand lasting for more than a few nanoseconds. Notably, this location was not where a binding pocket was expected [24,25,26]. A possible explanation for the divergent results is that the current model is not an accurate representation for the conditions in a spore. First, in the current modelling, water is in abundance whereas it is presumably scarce on the inner leaflet of the spore IM. There is also recent evidence from biochemical assays that Ca^2+^ in core CaDPA may be interacting with IM’s inner leaflet phospholipid head groups [40]. Therefore, transient interactions of L-alanine in the simulation may be of more significance in an environment with little water and CaDPA present on one side of the membrane [41].

Second, we have examined GerAB in isolation, but in the spore it most likely functions in a complex with GerAA and GerAC and in close proximity to GerD and other GRs [42]. These other GR-proteins may influence the structure of GerAB and thereby also the function. In such a complex, a potential scenario may be that GerAA binds L-alanine and undergoes a conformational change. This change would then alter the structure of GerAB such that it can receive L-alanine from GerAA [7,10,43], and subsequently transport it to the inner surface of the IM, instead of L-alanine inducing a conformational change in GerAB directly from the solvent. Third, differences in membrane properties such as membrane thickness or area per lipid can result in nonuniform surfaces of the membrane [44]. As observed by MD simulations of Aguayo et al. (2012), the non-uniformity of the surface of CL-containing bilayers may contribute to the interaction between the membrane and proteins, where CL mediates this interaction, like a scaffold, or lowers the fluidity of the membrane [44]. Both the tethering effect, facilitation of clustering between GR proteins, and the lower fluidity of a CL containing membrane could be of great influence for the germinosome and thus GerAB functioning [31].

GerAB itself seems to be structurally related to ApcT transporters [6]. Unlike most APC transporters, GerAB is presumably composed of 10 TMs instead of 12 based on our data. Here, the model selected as best fit contains 8 TMs instead of 10 as determined in this study. However, previous structural analysis of crystallized proteins suggested that the first 10 TMs are most important for GerAB function [24,38,45]. These two extra helices may affect the structure. Nonetheless, the structure of 3GI8-RaptorX, which contains 10 TM regions, is quite similar to 3GI8-SWISS suggesting that TM9 and TM10 do not affect the structure to a large extent. Earlier mutagenesis experiments suggest that TM1 and TM6 may play a critical role in ligand binding [6]. In addition, structural analysis studies of other APC proteins suggested that partial unwinding in the middle of TM6 increases the flexibility of TM6 or the entire protein and may be required for ligand binding [24,38]. In the current MD study the unwound region extended over the entire TM6 region.

Our results confirm that GerAB adopts a canonical APC superfamily fold in which TM1-TM5 resemble TM6-10 by a pseudo two-fold symmetry axis [26], as previously proposed by Cooper and Moir (2011) [6]. In addition, they [6] suggested a similarity between GerAB and ApcT as well as LeuT (PDB-code 2A65 [24]). Visual inspection of GerAB models also revealed large structural similarity with LeuT. Similar to LeuT, GerAB shows breaks in the helical structure of TM1 and TM6 halfway across the membrane. Since the structure of GerAB is very similar to LeuT, the L-alanine binding site may consist of partially unwound TM helices 1 and 6 [24]. Although no water channel has been observed in LeuT [24], GkApcT does contains a water filled channel similar to that in GerAB [26].

In contrast to ApcT transporters which are able to bind a ligand and then transport it across the membrane, mutagenesis, inhibitor and gene-silencing experiments suggest that GerAB functions more as a receptor instead of a transporter [18,23]. Yet, we revealed that GerAB contains a large water channel traversing the entire membrane spanning portion suggesting that there may be a role for GerAB as a transporter as well as a receptor.

Inducing germination in a laboratory setting often involves a mixture containing potassium implying that K${}^{+}$ is required to effectively induce germination [46]. On the contrary, LeuT requires 2 sodium ions as a co-ligand [24] while GkApcT and ApcT K158A are proton dependent for their transport [25,26]. Jungnickel et al. (2018) suggested that even though ApcT is a proton coupled transporter, a sodium ion might stabilize the unwound regions in TM1 in particular [26]. This may be the same for GerAB, but here instead of sodium, potassium is required. However, no such stabilization has been observed in the current research with either potassium or sodium as no interactions between potassium and the protein have been observed by visual inspection of the simulations.

Finally, there are a number of events in Bacillus spore germination triggered via GRs in which there is movement of small molecules across the spore IM. The first is the rapid release of much spore core H${}^{+}$, 85% of spore core Na${}^{+}$ and 75% of spore core K${}^{+}$ [47]. This event seems associated with a change in the IM and is accompanied by spores’ commitment to germinate [15]. The second event is the slow release of 15% of spores’ CaDPA pool in 10–15 min depending on the temperature. This slow release is then followed by the rapid release of all remaining CaDPA in 2 min [48]. The other molecule moving across the IM in germination is the water uptake in parallel with rapid CaDPA release, raising *B. subtilis* core water content from 35% of wet wt to 45%, and then from 45% to 80% when cortex peptidoglycan hydrolysis and core swelling complete spore germination [1]. None of the GR-dependent events above require ATP [2], and CaDPA release is most likely via the IM SpoVA protein channel [1,48]. How water enters the spore core, especially as all CaDPA is rapidly released, is not clear as an aquaporin by which water can cross plasma membranes of many bacteria, has not been identified in Bacillus species. How free monovalent cations are released from spores is also not known. However, there is likely a change in the IM at this time in germination, which is associated with spores becoming irreversibly committed to germinate [15]. The information above makes the water channel in GerAB identified in this work extremely attractive as playing a key role in early GR-dependent germination events. Indeed, a simple model is that there is a conformational change in GerAB triggered by germinant binding to the GR resulting in the formation of the water channel, and it is via this channel that monovalent cations are released, leading to commitment and subsequent germination events. One piece of evidence for the importance of this water channel in spore germination is that when E51, a GerAB residue lining the predicted water channel, is changed to leucine, the GerA GR no longer triggers spore germination [6]. In addition, the C subunit assembles normally in this GR variant, and likely also the A subunit [7]. Going forward, further mutagenesis of residues lining the water channel and examining effects of the mutant proteins on spore germination, including GR assembly, CaDPA and K${}^{+}$ release and commitment, as well as MD studies on the mutant proteins, would seem to have a very good chance of defining the role this water channel plays in spore germination.

In conclusion, most flexibility in structure and orientation was observed in TM1, TM6, EL1 and IL4. Interactions with water and the membrane were investigated which showed distinct regions that solely interact with water or membrane, but also some regions that exhibit a more mixed pattern in TM1 and TM6. Besides the interaction with water and membrane tails, the interaction with the potential ligand L-alanine was also examined, and this found that L-alanine is often associated with residues in intra- or extracellular loops (EL3 and IL3). Finally, this qualitative study reveals that GerAB contains a water channel traversing the membrane spanning portion of the protein. Unlike predicted by TMHMM/Phobius, our results show that GerAB does not contain 11 TMs but most likely 10. The predicted TM helix 8 is in fact IL4 instead of a TM domain. Prior mutagenesis studies by Cooper and Moir (2011) showed a number of residues that affected germination when mutated [6]. We have shown that these residues are probably part of TM1, TM2, TM6, and TM8. Moreover, our results show that L-alanine is close to both P178 and R271 in agreement with previous results from Cooper and Moir (2011) [6].

The results in this study may inspire further mutagenesis studies. In particular, it would be interesting to study TM1 and TM6 in more detail. In addition, this study provides predictions on which residues may be involved in interaction between GR-proteins (e.g., residues of TM1 and 6 lining the channel). Furthermore, the effect of Ca^2+^ and Mg^2+^ on the CL membrane would be an interesting topic to investigate, as these ions are known to strongly modulate lipid properties [49] and therefore might affect GerAB functioning. Mutagenesis studies in vitro could also examine a CL deficient *B. subtilis* to study the effect of this membrane component on protein structure.

## 4. Materials and Methods

### 4.1. Starting Structure

Protein sequences were obtained from Uniprot [36] entry UniProtKB - P07869 (GERAB_BACSU) and homology-based models were created using SWISS-Model and RaptorX [33,35]. For the latter we used the template-based approach. In order to construct the GerAB model, the structure from ApcT transporters with pdb code 3GI8 [25] and pdb code 6F34 [26] were used as templates, see Figure 1 for the sequence alignments used to construct the model. Using the web-based server CHARMM-GUI [50,51,52,53], proteins were inserted by the replacement method in a lipid bilayer composed of 100 × 100 (numbers of lipid molecules) 4:1 POPG: POPE and 6:3:1 POPG: TMCL2: POPE in a Charmm36 forcefield [44,54,55]. Ligand structures from zwitterion amino acids were obtained from ChEBI [56] and converted to mol2 files using Avogrado (version 1.2.0). With CGenFF a .str file was created [57,58]. Water molecules and ions in the models obtained from CHARMM-GUI were removed prior to passing it on to MD simulation.

### 4.2. Molecular Dynamics Simulations

All simulations were performed in Gromacs 4.5.5 using the Charmm36 forcefield extended to include TMCL2 parameters [44]. TIP3P water was used to describe the water molecules. After combining the topology and coordinate files of the ligands and GerAB embedded in the membrane, a box was formed, 1 nm smaller in the X and Y direction to prevent lipids from laying parallel. The box was solvated with TIP3 water molecules. Water molecules placed in the lipid bilayer were removed after which the system was neutralized by adding 0.02 M KCL. Table 1 lists the systems used in this work. Long-range electrostatic interactions were treated using the Particle Mesh Ewald method [59] with a grid spacing smaller than 1 Å and for short-range non-bonded interactions the cutoff value was set on 12 Å. Energy minimizations of the solvent were performed with the steepest descent integrator and followed by 100 ps heating under NVT conditions using V-rescale thermostat to reach 278 K. While the protein was temperature coupled with the membrane molecules, the ligand was coupled to the solute and ions. In the NPT ensemble, the pressure was controlled with semiisotropic pressure coupling using the Parrinello-Rahman barostat. The reference pressure was set at 1 bar. The MD simulations were performed at 298K with a time step of 2 fs. H-bonds and bond lengths were restrained using Lincs and periodic boundary conditions were applied to all MD simulations. MD ensembles were simulated for 100 ns. Five runs were performed for each model (6F34, SWISS, 3GI8-SWISS and 3GI8-RaptorX, each with starting velocities randomly drawn from the Maxwell-Boltzmann distribution, followed by 1 ns equilibration) and multiple controls were included, for which each 1 run was performed (Table A1). The temperature was set at 298 K and every 10 ps frames were saved. Prior to analysing the data, the first 50 ns of the simulations were removed. Structural analysis is based on the last frame of each simulation since the models still showed some flexibility at the end of the simulation (See Figure A1 for RMSD plots.)

### 4.3. Analysis

To analyze the structure of GerAB we calculated the distance between the hydrogen bond acceptor C=O in residue n and the hydrogen bond donor in N-H in residue n + 4. Then these distances were averaged over the last 50 ns of each of the 5 runs for each system. The interactions of GerAB with the other components in the model were determined by computing the minimum distance and counting the number of times the distances dropped under 0.4 nm. The probability of finding a component within 0.4 nm of a protein residue is labelled as proximity. Proximity was calculated for water, membrane tails, L-alanine and potassium ions. In addition, we computed the number of binding events, and the average residence time, i.e. the time each binding event lasted. A binding event is defined as the first frame dropping below the interaction cutoff distance of 0.4 nm. The average time per event was computed by counting the number of frames following an event for which the distance to the residue was below the interaction cutoff. Finally, we computed the dimensions of the water channel by using the HOLE software package, using a maximum radius of 1 nm [37].

All analysis was performed using GROMACS 4.5.5 tools [60] and PyMol 2.3.4 was used to visualize the protein structures and create images. Plots were created with the Python library matplotlib.

## Figures and Tables

**Figure 1 ijms-22-03793-f001:**
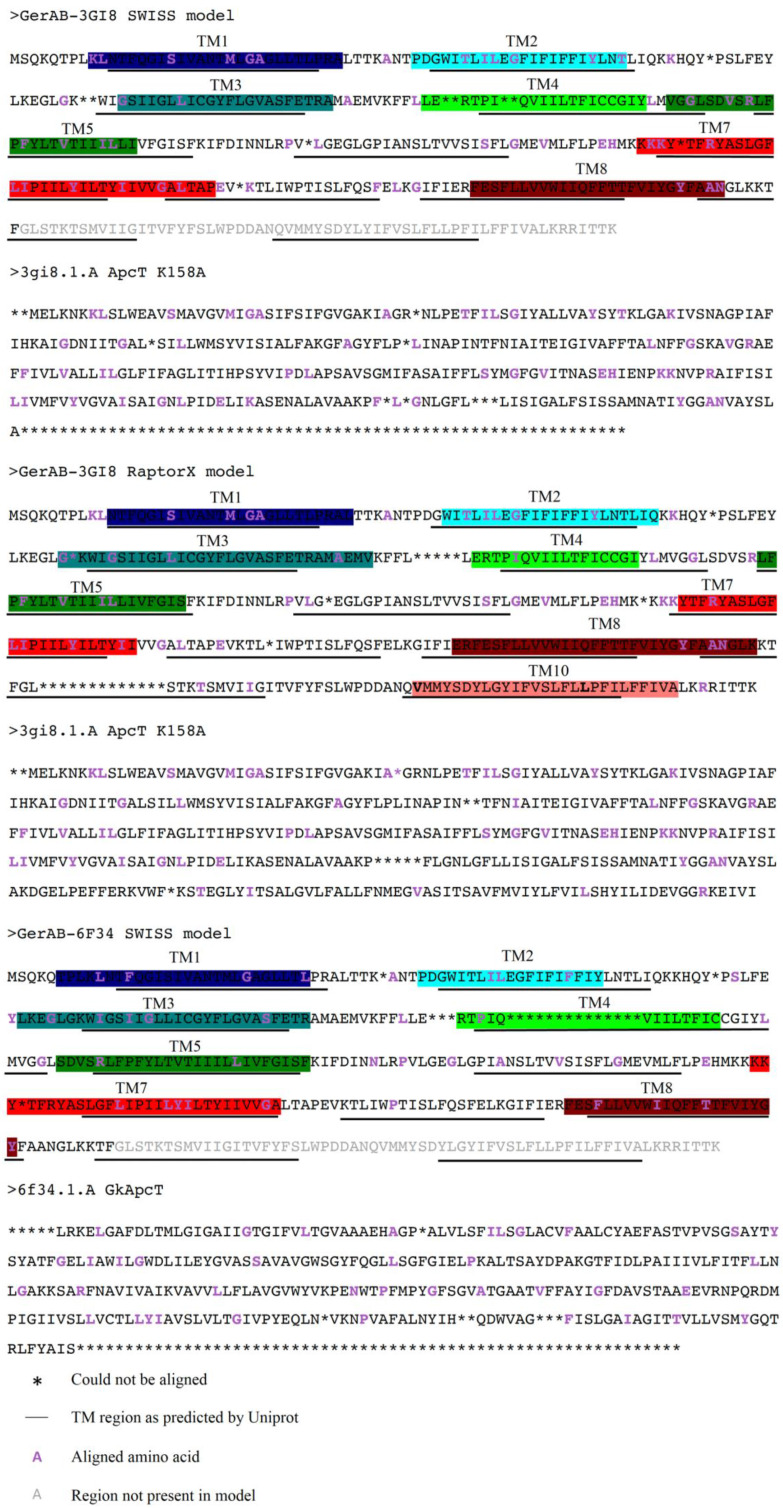
Alignment of GerAB models with templates. GerAB was aligned with 3GI8 ApcT K158A (pdb code 3GI8, [25]) and GkApcT (pdb code 6F34 [26]) [33,35]. For 3GI8-SWISS as well as 6F34-SWISS, 303 amino acids were aligned. For 3GI8-RaptorX 365 amino acids were aligned. Identical amino acids are displayed in purple font, transmembrane (TM) domains as established in this research are highlighted in color, corresponding to Figure 2 and TM domains as currently defined by TMHMM/Phobius are underlined [19,20,21,22]. Asterisks indicate that the residues could not be aligned and gray letters on the C-terminus indicate the domains that are not present in the models.

**Figure 2 ijms-22-03793-f002:**
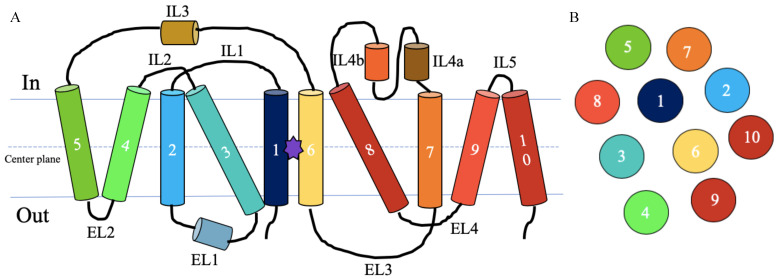
Schematic overview of GerAB topology. (**A**) This topology prediction is based on the structures of the three GerAB models examined in this study. The helices are shown as cylinders and the possible L-alanine binding pocket is depicted as a purple star between TM1 and 6. The figure is adapted from [24]. (**B**) illustrates the organisation of the protein from the inside of the IM. Colors of the helices depicted as circles correspond to A.

**Figure 3 ijms-22-03793-f003:**
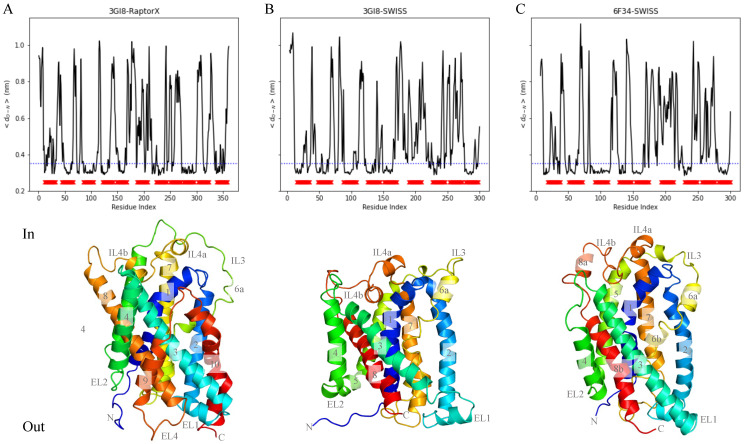
Structure of GerAB models. (**A**) shows the helicity and structure of the 3GI8-RaptorX model [25], with the colors indicating the helical regions as defined in Figure 2. (**B**) depicts the helicity and structure of the 3GI8-SWISS model [25]. (**C**) shows the helicity and structure of 6F34-SWISS model [26]. The helicity is defined as the distance between the backbone oxygen atom of residue n and the backbone nitrogen atom of residue n+4. The blue line indicates a distance of 0.35 nm, the distance threshold for a hydrogen bond interaction and the red blocks indicate TM regions as defined by TMHMM/Phobius, see Uniprot entry UniProtKB-P07869 (GERAB_BACSU) [36]. Helicity per model is the average of the last 50 ns of 5 runs.

**Figure 4 ijms-22-03793-f004:**
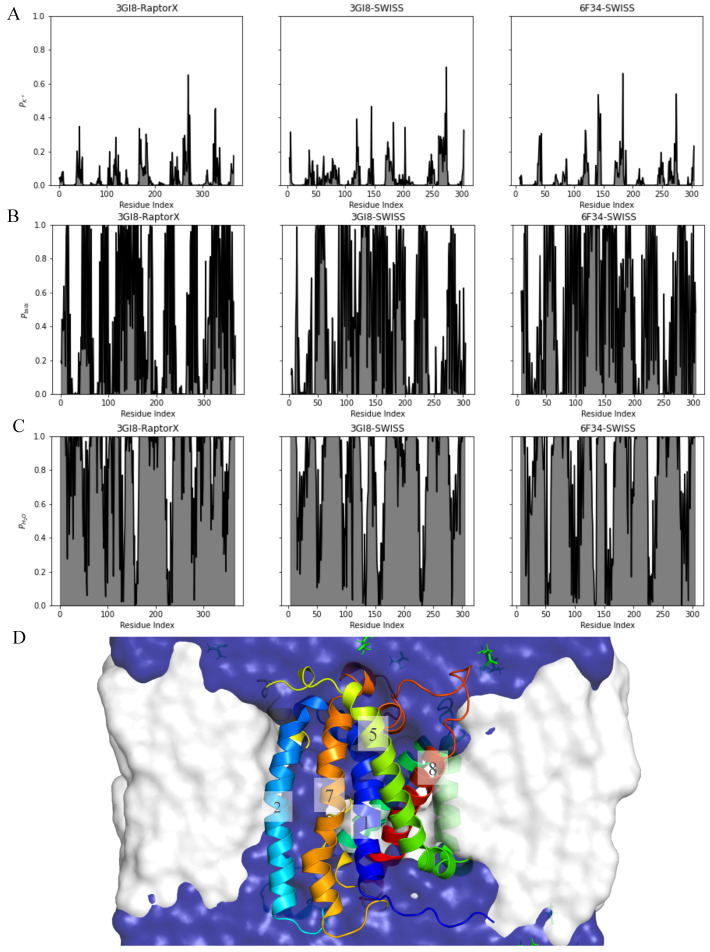
Interaction of GerAB residues with potassium, membrane tails and water. (**A**–**C**) shows the proximity plots for K^+^, membrane tails and water to residues of GerAB. A P value of 1 indicates that water or membrane tails are within 0.6 nm of the residue for the duration of the simulation, while a value of 0 indicates that water or lipid tails are never within a distance of 0.6 nm of the residue. The proximity is calculated over 5 runs for each model. (**D**) shows a snapshot of the complex with the membrane in white, water in blue and GerAB helices in color, corresponding to Figure 2.

**Figure 5 ijms-22-03793-f005:**
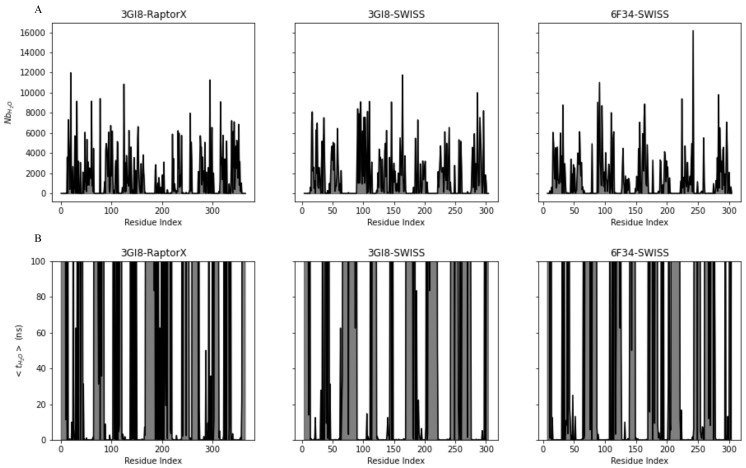
Number of binding events and average residence time for water to GerAB residues. In (**A**) the average number of binding events (Nb) are shown for residues in 3GI8-RaptorX, 3GI8-SWISS and 6F34-SWISS. (**B**) shows the average residence time of water to residues in 3GI8-RaptorX, 3GI8-SWISS and 6F34-SWISS.

**Figure 6 ijms-22-03793-f006:**
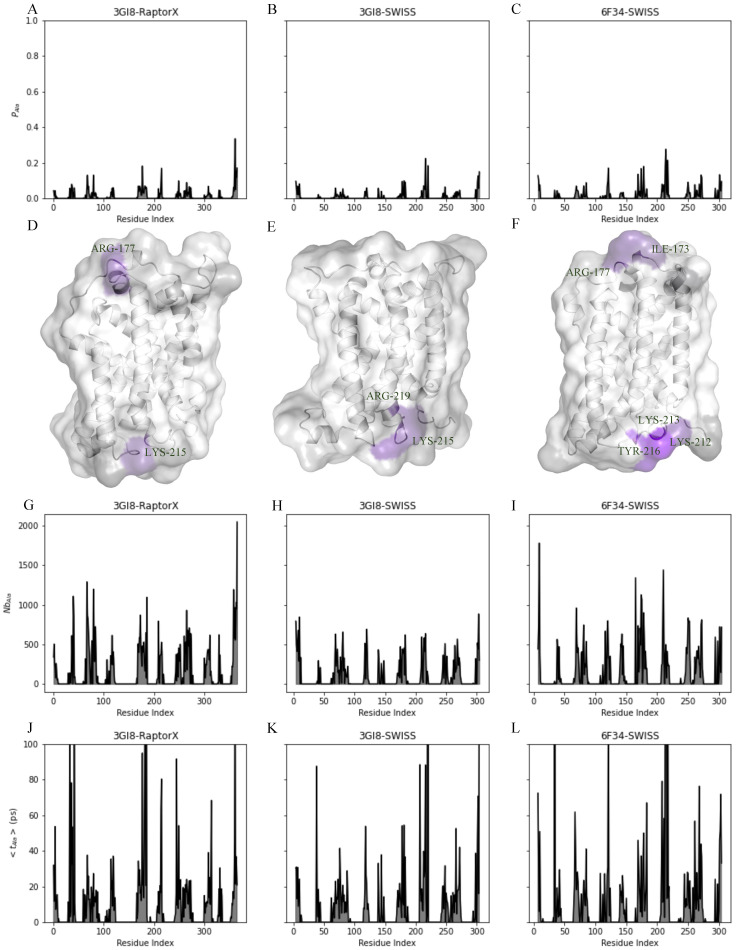
Interactions of GerAB with L-alanine. (**A**–**C**) shows the average proximity of L-alanine to each residue in the three GerAB models. Regions with higher proximity can be identified and are colored purple in the structures (**D**–**F**) Graphs (**G**–**I**) show the number of binding events of L-alanine to residues in the GerAB models. (**J**–**L**) show the average residence time of L-alanine to residues in the GerAB models.

**Table 1 ijms-22-03793-t001:** System details. n indicates the number of runs per system.

Systems	n	Time (ns)
POPG-TMCL2-POPE membrane (6:3:1)		
6F34-SWISS, 20 L-ala, 0.02 M KCL, 298 K	5	100
3GI8-RaptorX, 20 L-ala, 0.02 M KCL, 298 K	5	100
3GI8-SWISS, 20 L-ala, 0.02 M KCL, 298 K	5	100
3GI8-SWISS, no ligand, 0.02 M KCL, 298 K	1	100
3GI8-SWISS, 1 L-ala, 0.02 M KCL, 298 K	1	100
3GI8-SWISS, 20 L-ala, 0.02 M NaCL, 298 K	1	100
3GI8-SWISS, 20 L-ala, 0.02 M KCL, 310 K	1	100
3GI8-SWISS, 1 D-ala, 0.02 M KCL, 298 K	1	99
3GI8-SWISS, 1 L-val, 0.02 M KCL, 298 K	1	99
Membrane only, 0.02 M KCL, 298 K	1	100
POPG-POPE membrane (6:4)		
3GI8-SWISS, no ligand, 0.02 M KCL, 298 K	1	50
3GI8-SWISS, 20 L-ala, 0.02 M KCL, 298 K	3	100

## Data Availability

Input files for the MD simulations reported here are openly available at FigShare, 515doi:10.21942/uva.13655516.v4. provided at https://uvaauas.figshare.com/articles/dataset/GerAB/13655516 (1 March 2021).

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
