# Peer review of "Predicting the Structure and Dynamics of Membrane Protein GerAB from Bacillus subtilis"

_ijms, 2021, doi:10.3390/ijms22073793_

Round 1

Reviewer 1 Report

The paper by Blinker et al. studies the structural dynamics of membrane protein, germinant receptors (GRs) from Bacillus subtilis. GRs respond to ligands (e. g. L-alanine) and are involved in triggering spore germination. In B. subtilis consists of three subunits GerAA, GerAB, and GerAC, however there is a lack of structural information. To fill this gap the authors performed homology modelling with MD simulations. They predicted that GerAB is a multi-spanning transmembrane protein containing a water channel. Moreover they pointed out the specific binding sites of L-alanine on GerAB. Although, taking into consideration the size of the system, the MD runs are relatively short (up to 100ns), the authors clearly state they are not converged and cannot be interpreted quantitatively. In the opinion of the Reviewer, the article provides important structural insights towards the molecular mechanism of spore germination.

Author Response

We would like to thank the reviewer for the positive views regarding our manuscript. Indeed our results are to be interpreted in terms of gerAB functioning more qualitatively. The data presents a first study using homology modelling and molecular dynamics (MD) to provide structural predictions for the spore inner membrane located protein GerAB. Our predictions indicate that GerAB is a helical transmembrane protein that contains a water channel. MD simulations in the presence of free germinant L-alanine show that alanine binds transiently to specific sites on GerAB. Together the data provide a theoretical framework for further experimental study of early molecular mechanistic events in bacterial spore germination.

Reviewer 2 Report

Blinker et al. predicted an atomic model of Bs GerAB and carried out molecular dynamics (MD) simulations of it. This protein is a part of a large protein complex called germinosome responsible for germination and outgrowth of bacterial spores. While several components of this complex have been structurally characterized before, there is no high-resolution structure of GerAB available to date. Since spore germination is responsible for such important processes as food spoilage, obtaining detailed structural information about the molecular machinery responsible for it is highly demanded.

However, in the reviewer’s opinion, the provided models should be properly validated before any rigorous conclusions can be made based on them. Such validation is currently missing what precludes publication of this manuscript in its present form.

Specific points:

  1. From the text (line 105), it is not clear, why only two particular templates were chosen for homology modeling (i.e., 3GI8 and 6F34) while a number of other potential templates were ignored, such as 5OQT, 3GIA, 7KGV. All of these templates have similar similarity % to the target sequence. The sequence similarity and identity are key factors defining the potential quality of the built homology models (as a rule of thumb, similarity must be over 20-30% to allow adequate models for the proteins of similar size). These metrics should be provided.
  2. The quality of the obtained models must be evaluated and compared between each other using standard metrics in the field, such as DOPE, GA341, SOAP (see https://salilab.org/bioinformatics_resources.html for available software).
  3. Since the available templates do not cover the full target sequence, the authors used de novo protein structure prediction by the RaptorX algorithm as an alternative approach. It is not clear, however, which particular version of the algorithm was used: the new one relying on the statistical inference of amino acid contacts from the multiple sequence alignment, or the older one, which utilizes a template-based approach.
  4. Apart from the quality estimation of all the built structural models, they should also be compared with each other in terms of the pairwise backbone RMSD to see how structurally different they are (line 153). These results altogether should allow picking the best model for further investigations in a more rigorous way.
  5. The behavior of the protein in the presence of one and multiple L-Ala molecules, as well as the overall stability of the simulated systems, must be checked by tracking the protein RMSD over the simulation time not just by visual inspection (line 120-127).
  6. It would be beneficial to check if the obtained models satisfy the positive-inside, negative-outside rule for the topology of transmembrane proteins.
  7. Line 150: “the loop IL3 contains a helix of 1-2 amino acid long”. That is an inaccurate phrase as 1-2 amino acids are less than a single turn of an a-helix (~3.6 residues/turn) so it is not possible to reliably define the helicity of such a short fragment.
  8. Throughout the manuscript, the authors use a quite uncommon definition of helicity based solely on the distance between a putative H-bond donor (N-H) and acceptor (C=O) in the protein backbone. The H-bond formation depends on the angle between an acceptor and donor apart from the distance. Moreover, there is a number of convenient standard tools for the assignment of the secondary structure, such as DSSP and STRIDE. It is highly recommended to use them.
  9. When the authors compare the transmembrane (TM) helices observed in their structural models with the predicted TM helices, they refer to Uniprot. It is not correct, as Uniprot itself does not predict secondary structure and protein TM topology but relies on TMHMM and Phobius algorithms. These original programs shall be referenced properly. The reviewer would also strongly suggest providing a more comprehensive comparison of predictions made by various secondary-structure/TM helix prediction tools (Phobius, TMHMM, MemBrain, TMpred, etc., and their atomic models.
  10. Line 204-205: “…since this model deviates the least from its template structure and contains the most intact helices as determined by the helicity.” Such criteria for the selection of the best model are not obvious for the reviewer. Why the least deviation from the template guaranties that the chosen model is the most optimal one?
  11. Line 215: “The regions that are in close proximity to water are often IL or EL regions”. It’s a self-evident fact that extracellular and intracellular loops do form contacts to water.
  12. At the same time, the phrase in lines 219-220, “less hydration means more helical structure” is not a general rule and needs either further justification or shall be removed.
  13. It is important to include RSMD over time plots for all obtained MD trajectories as without them it is impossible to judge about convergence or at least sufficient equilibration of the molecular systems. In turn, the latter is a prerequisite to draw any conclusions about the formation of a water-filled pore, which can be just an artifact of insufficient equilibration.
  14. Line 245: the authors use visual inspection to track the formation of a water pore. It would be more accurate to describe it in a quantitative way, e.g., by means of the HOLE tool (http://www.holeprogram.org/) or similar tool.
  15. Figure 5: why only these residues were considered as those lining the pore? More accurate profiling of a putative pore (see comment 14) should allow selecting residues forming a pore more precisely.
  16. Why have the authors preferred their setup for the determination of the L-Ala binding site(s) over docking? Given relatively short simulation time (less than 300 ns in total), their approach may suffer from insufficient sampling. Correspondingly, the identified binding poses might represent not all of the possible poses/not those with the lowest energy.
  17. Lines 297-8: “This is deduced from the GerAB-RaptorX model, but most results are about the GerAB-3GI8 model.” If the most of results (incl. MD simulations) were obtained for the GerAB-3GI8 model, why the authors refer to the GerAB-RaptorX model, which is considered by them as a non-optimal one?
  18. Line 336: “Our structural model for GerAB shows great structural similarities with ApcT transporters.” In the reviewer’s opinion, it’s not surprising that a structural model is similar to its template. Perhaps, it’s meant that GerAB itself (not a built model) seems to be structurally related to ApcT transporters what allowed the authors to use them as a template for homology modeling.
  19. Line 423: The units are missing for the lipid bilayer size. 100x100 Angstrom?
  20. Line 433: “…to create asymmetrical solvation”. It is not clear what the authors mean given that they state that the periodic boundary conditions were used in all simulations.

Reviewer 3 Report

In this contribution by Blinker et al. the possible topology and three-dimensional structure of an important germination tranceptor / transporter GerAB from B.subtilis are presented and possible interactions with its substrate L-Ala are calculated and discussed.  The topic is interesting and important for the field but there is too little novelty in my opinion. This originates from the pure in silico nature of this manuscript - which does not mean that the presented calculations are not reliable or wrong, this is indeed very powerful technique, but in the current context - what do we learn? - that we cannot even reliably predict the number of trans membrane helices? As reported by authors (lines 153-166) three calculated models differ not only in the details, but even in the number of transmembrane helices! 

Furthermore, why did authors not consider generating a model with the evolutionary couplings method? That would perhaps help to determine the best model. 

It was also disappointing that the actual binding site of L-Ala was not found, and the present results indicate only unspecific binding. Perhaps authors should compare their results with the down binding sites for L-Ala. 

I think the most interesting part of this manuscript is the revealed water channel, so perhaps more focus should be on this part. Is it forming a continuous pore? If so, can L-Ala just go along? Or is the role of L-Ala only induce the water channel opening? Why not to run the simulation of monovalent ions (it is not that difficult as there are good force field parameters for monovalent ions) to make the conclusions stronger? Also in silico mutagenesis would be a nice addition to the manuscript.

The authors discuss that perhaps their conditions are not matching the spore conditions (lines 316-320) - but why not to simulate it? that's the beauty of the computational biology that everything can be done in silico without the need for the tedious lab experiment.  Again, if the complex formation is needed - why don't you run calculations on the complex, especially since there is structural information available on other subunits? 

In summary I think the manuscript touches on a very interesting topic, but gives too little to a reader at the end. Running additional simulations would make this manuscript way more attractive. 

Round 2

Reviewer 2 Report

The presentation of data and the analysis have been significantly improved in the revised manuscript so it can be now published in IJMS.

Reviewer 3 Report

I thank the authors for the careful revision and addressing all points which I raised. I recommend their revised work for publication